# Stress Hyperglycemia as a Prognostic Indicator of the Clinical Outcomes in Patients with Stroke: A Comprehensive Literature Review

**DOI:** 10.3390/biomedicines13081834

**Published:** 2025-07-28

**Authors:** Majed Mohammad Alabdali, Abdulrahim Saleh Alrasheed, Fatimah Ahmed Alghirash, Taif Mansour Almaqboul, Ali Alhashim, Danah Tareq Aljaafari, Mustafa Ahmed Alqarni

**Affiliations:** 1Neurology Department, College of Medicine, Imam Abdulrahman Bin Faisal University, Khobar 34445, Saudi Arabia; mmalabdali@iau.edu.sa (M.M.A.); ahalhashem@iau.edu.sa (A.A.); dtaljaafari@iau.edu.sa (D.T.A.); mqarni@iau.edu.sa (M.A.A.); 2Department of Neurosurgery, College of Medicine, King Faisal University, AlAhsa 31982, Saudi Arabia; 3College of Medicine, King Faisal University, AlAhsa 31982, Saudi Arabia; 219013169@student.kfu.edu.sa; 4College of Medicine, Batterjee Medical College, Jeddah 21442, Saudi Arabia; 150341.taif@bmc.edu.sa

**Keywords:** stroke, diabetes mellitus, hyperglycemia, prognosis, review

## Abstract

**Background:** Stress hyperglycemia (SH), a transient elevation in blood glucose levels during acute stress such as stroke, has been increasingly recognized as a critical determinant of clinical outcomes. This review aims to evaluate the association between SH and clinical outcomes across different stroke subtypes and its role as a prognostic indicator. **Methods:** The current literature review was conducted through a comprehensive literature search of PubMed, Scopus, and Web of Science electronic databases. Initial title and abstract screening was conducted by two independent reviewers depending on the relevance to the topic of interest. Final study inclusion was based on the clinical relevance and agreement between reviewers. **Results:** Current evidence links SH with higher stroke severity (Higher national institutes of health stroke scale (NIHSS)), larger infarct volumes, increased risk of hemorrhagic transformation, and worse functional recovery (Lower modified rankin scale (mRS)), especially in ischemic stroke. In hemorrhagic stroke, SH is associated with hematoma expansion, perihematomal edema, and worsening neurological function. Although SH has been shown to be a reliable stroke outcome predictor, there is no scientific consensus regarding the most reliable measurement method. The use of absolute blood glucose values may not accurately reflect SH, particularly in diabetic patients, where chronic baseline hyperglycemia complicates interpretation. This underscores the necessity for individualized assessment rather than a uniform interpretation. Clinically, the early detection of SH may provide enhanced monitoring and supportive care; however, rigorous glucose management remains contentious due to the risk of hypoglycemia. **Conclusions:** This review synthesizes evidence from recent studies and supports SH as a prognostic marker of both short- and long-term adverse outcomes in stroke patients. Further research is warranted to evaluate the efficacy of targeted glycemic treatments on such outcomes.

## 1. Introduction

Stroke is a medical emergency caused by an interruption or reduction in blood supply to the brain, leading to tissue damage and neurological deficits [1]. It is classified into two main types: ischemic stroke, which results from a blocked artery due to a thrombus or embolus, and hemorrhagic stroke, which is caused by the rupture of a blood vessel [1]. Common signs and symptoms include sudden weakness or numbness, particularly on one side of the body; difficulty speaking or understanding speech; vision disturbances; loss of coordination; and severe headache [2].

Globally, stroke is still the second leading cause of death and the third leading cause of death and disability combined, based on the latest global burden of disease (GBD) 2021 stroke burden estimates [3]. Between 1990 and 2021, the global stroke burden increased markedly, with incident strokes rising by 70%, stroke-related deaths by 44%, prevalent stroke cases by 86%, and disability-adjusted life years (DALYs) by 32%. Notably, most of this burden, 87% of deaths and 89% of DALYs, occurred in low-income and lower-middle-income countries [4]. Stroke-related DALYs have continued to rise in absolute numbers, largely driven by sedentary lifestyles, representing a current and potentially growing threat to healthcare systems in the coming decades [4]. These data highlight that stroke not only causes high mortality but also leads to significant long-term disability, including motor, cognitive, speech, and psychological impairments [2]. The high rates of disability and the need for ongoing care and rehabilitation reflect the complexity of stroke recovery, highlighting the importance of reliable prognostic indicators to predict and guide treatment and improve outcomes for stroke patients.

One such potential predictor is stress hyperglycemia (SH), also known as stress-induced hyperglycemia, which is a transient elevation in blood glucose levels during acute illness, commonly observed in patients without prior diabetes mellitus (DM) [5,6]. It represents a primary neuroendocrine response to physiological stress, commonly seen in critically ill patients, including those with stroke, sepsis, major trauma, and myocardial infarction [7,8]. The acute elevation in blood glucose observed during stroke is increasingly recognized not just as a biochemical response but as a significant prognostic indicator linked to unfavorable clinical outcomes [9]. Several studies have found SH to independently predict adverse short-term outcomes, such as early neurological deterioration (END), increased infarct size, and higher rates of hemorrhagic transformation [10,11]. Notably, SH has also been significantly linked to worse 3-month outcomes, including a twofold risk in functional decline, a 63% higher rate of functional dependence, and nearly double the risk of mortality [12]. However, its precise prognostic value remains controversial, as a study by Osei et al. (2017) failed to find consistent associations between SH and long-term functional outcomes, suggesting that SH may reflect stroke severity rather than being an independent prognostic factor [13]. These discrepancies challenge the use of SH as a prognostic marker and underscore the complexity of glucose regulation in acute stroke. Moreover, differences in the prognostic value of SH have been observed between diabetic and non-diabetic patients. In one study, after stroke, non-diabetic patients with SH had a 1.7-fold higher risk of in-hospital 28-day mortality than diabetic patients [14].

In addition, admission hyperglycemia is linked to worse outcomes and higher in-hospital mortality in both ischemic and hemorrhagic strokes, though DM appears to raise mortality risk only in hemorrhagic stroke, highlighting a potential difference in the pathophysiological mechanisms between the two subtypes [15]. The inconsistencies across studies may result from heterogeneity in SH definitions, glucose cutoff values, and timing of measurement, highlighting the need for standardized measures to enhance its reliability as a prognostic tool in stroke.

Many earlier studies used only fasting or admission glucose to define SH, without accounting for pre-stroke glycemic status. For instance, admission glucose levels exceeding 210.5 mg/dL in diabetic patients and 113.5 mg/dL in non-diabetic patients were linked to significantly higher in-hospital mortality [15]. This limitation may explain why hyperglycemia often fails to predict outcomes in diabetic patients with chronically elevated glucose that may mask the prognostic value of acute glucose elevations. Thus, newer metrics such as the stress hyperglycemia ratio (SHR), glycemic gap (GG), and glucose-to-glycated hemoglobin (HbA1c) ratio (GAR) have been introduced. These measures combine acute glucose readings with HbA1c, which reflects average blood glucose over the preceding 2–3 months [16,17,18]. This approach, known as relative hyperglycemia, compares acute glucose levels to a patient’s usual glycemic background, allowing for a more individualized assessment of glucose dysregulation [9]. Sun et al. (2023) reported that an SHR ≥ 0.89 and a GG ≤ −0.53 were significantly associated with poor 90-day outcomes and an increased risk of hemorrhagic transformation in ischemic stroke patients undergoing mechanical thrombectomy (MT), supporting the prognostic utility of integrated glycemic metrics over absolute glucose thresholds [17].

Despite growing evidence, there is still no scientific consensus on the most reliable method for predicting outcomes related to SH in stroke. The variability in assessment tools and cutoff values complicates the interpretation of SH’s prognostic significance. Moreover, there remains a lack of comprehensive reviews that consolidate these prognostic findings and evaluate the different methods of SH measurement in both ischemic and hemorrhagic stroke. Therefore, this study aims to provide a detailed overview of SH in stroke, focusing on its implications for prognosis and therapeutic outcomes and highlighting the need for standardized predictive approaches.

## 2. Methods

A comprehensive literature search was carried out across PubMed, Scopus, and Web of Science databases. The used relevant search strategy was ((Stress Hyperglycemia[Mesh] OR Acute Hyperglycemia OR Transient Hyperglycemia OR Hyperglycemia) AND (Stroke[Mesh] OR Ischemic Stroke OR Hemorrhagic Stroke OR Cerebrovascular Accident) AND (Prognosis[Mesh] OR Outcome OR Mortality OR Functional Recovery OR Neurological Outcome OR Disability)). The search covered all of the available literature up to January 2025, with no restrictions on study design or setting. However, language restrictions were applied, as only English-language studies were included.

Studies were included in this review if they involved patients with ischemic or hemorrhagic stroke, assessed the prognostic value of SH or related indices, and reported relevant clinical outcomes spanning the definition of SH through the management options. Additionally, studies were required to use validated definitions or measurements of SH and stroke outcomes. Exclusion criteria included studies that did not define SH clearly, lacked relevant outcomes, were not published in English, had no available full text, or were non-peer-reviewed articles such as editorials, letters, or abstracts.

All search results were imported into Rayyan software [19], and the duplicates were removed. Title and abstract screening was performed independently by two reviewers, followed by full-text evaluation for eligibility. Discrepancies were resolved through discussion. Data extraction was performed by two reviewers using a predesigned form, which recorded study details (Such as the first author’s name, year of publication, aims, study design, and study setting), patient characteristics (Such as type of stroke, either ischemic or hemorrhagic, and DM status), SH parameters (Such as the definition or measurement method, and the cutoff value), and outcome measures (Including short-term and long-term outcomes). Study tables were created using Microsoft Excel [20] and Procreate software (v5.3.15) [21] was used to illustrate all the figures. After following the inclusion and exclusion criteria, 142 publications were included in the final qualitative synthesis (Figure 1).

## 3. SH Prevalence and Risk Factors

SH is a common metabolic response in acute stroke, with an incidence of 24% in ischemic stroke patients. The incidence varies geographically, affecting 33% of ischemic stroke patients in North America, 25% in Europe, and 21% in Asia. Notably, patients with hemorrhagic stroke were reported to have even higher blood glucose levels compared to those with ischemic stroke [22].

Stroke severity is an important factor in the development of SH, as larger infarcts, particularly those resulting from large vessel occlusions, are commonly linked to higher blood glucose levels and worse clinical outcomes [23,24]. The stroke site also contributes to SH, as it is more common in lesions near the brain’s midline, particularly in the brainstem, due to greater activation of the hypothalamic–pituitary–adrenal (HPA) axis, sympathetic pathways, and release of glycotropic hormones [22]. Moreover, the presence of certain comorbidities at the time of stroke incidence can substantially raise the risk of developing SH. Hypertension (HTN), for instance, often leads to endothelial dysfunction, vascular stiffness, and end-organ damage and is commonly linked to metabolic syndrome, all of which are factors that worsen the harmful effects of SH on patient survival [25]. The relationship between DM and SH in stroke is complex. While SH is more frequent in non-diabetic patients following stroke and is generally associated with a higher relative risk of adverse outcomes [14], the superimposition of SH in stroke patients with pre-existing DM has also been linked to increased in-hospital mortality [18]. However, Ma et al. (2024) [26] found that HTN, more than DM, influences the predictive value of SH, possibly because diabetic patients experience chronic inflammation and oxidative stress. Furthermore, insulin therapy in diabetic patients may offer anti-inflammatory benefits, thereby improving outcomes and influencing SH predictive value [26]. Sharma et al. (2017) reported the frequency of SH as 24.13% in stroke, 19.54% in multiple organ dysfunction syndrome, 17.24% in chronic kidney disease, and 8.05% in chronic liver disease, all of which were significantly associated with hospital stays longer than 7 days [27]. The presence of these combined morbidities can further increase the risk of SH and may affect the prognostic significance of SH in such patients.

## 4. Overview of SH Pathophysiology in Stroke

SH in acute stroke results from complex interactions between neurohormonal, metabolic, and inflammatory mechanisms, which disturb glucose homeostasis [28]. The regulation of glucose metabolism is primarily controlled by the glucose regulatory centers in the brain, including the hypothalamus, brainstem, and cerebral cortex. These centers integrate signals from peripheral glucose levels, neurohormonal inputs, and energy demands, ensuring appropriate glucose availability. The glucose regulatory center, primarily located in the hypothalamus, plays a crucial role in maintaining glucose homeostasis. The ventromedial hypothalamus (VMH) acts as a key metabolic sensor by detecting glucose fluctuations and modulating insulin sensitivity and hepatic glucose output [29]. In stroke, stress signals activate the VMH, leading to excessive glucose production via stimulation of the HPA axis and sympathetic nervous system [30]. The nucleus tractus solitarius (NTS) in the brainstem also contributes by integrating autonomic inputs and amplifying sympathetic output, further enhancing hepatic gluconeogenesis. Additionally, the prefrontal cortex and amygdala influence stress responses, modifying glucose metabolism through their interactions with the hypothalamus. The imbalance in these glucose-regulatory networks results in increased circulating glucose levels, contributing to metabolic dysregulation in stroke patients [29]. The HPA axis and the sympathetic adrenomedullary system are key regulators of stress responses, contributing to hyperglycemia through the release of counter-regulatory hormones such as cortisol, epinephrine, norepinephrine, glucagon, and growth hormone, culminating in hepatic glycogenolysis and gluconeogenesis [31]. Additionally, humoral factors, including pro-inflammatory cytokines, such as tumor necrosis factor-alpha (TNF-α), interleukin-6 (IL-6), and interleukin-1 beta (IL-1β), further exacerbate hyperglycemia by impairing insulin signaling and increasing hepatic glucose output. These mechanisms, while initially adaptive in providing energy to vital organs, become detrimental in stroke, where excessive glucose availability promotes oxidative stress, excitotoxicity, and poor neurological outcomes [31,32] (Figure 2).

## 5. Pathophysiological Variations in SH in Ischemic Versus Hemorrhagic Stroke

The SH induction mechanism differs in ischemic and hemorrhagic stroke due to distinct pathophysiological processes. In ischemic stroke, cerebral hypoxia triggers a surge in counter-regulatory hormones, leading to excessive glucose production and impaired glucose utilization [33]. The brain shifts to anaerobic glycolysis, increasing lactate production and acidosis, exacerbating neuronal injury. In contrast, hemorrhagic stroke, particularly intracerebral hemorrhage (ICH), induces SH through elevated intracranial pressure, blood–brain barrier (BBB) disruption, and neuroinflammation [34,35]. The extravasated blood provokes an intense inflammatory response, often causing more severe SH than in ischemic stroke [36]. There are significant ramifications for acute care from this mechanical difference (Figure 3). Hemorrhagic stroke SH usually presents with more aggressive secondary brain damage. Timely glucose monitoring and control are therefore vital, particularly in ICH. The American heart association (AHA) advises acute stroke patients to keep glucose levels between 140 and 180 mg/dL to prevent problems from both hyper- and hypoglycemia and minimize unfavorable outcomes [37]. Knowing these mechanical and clinical variations helps guide the creation of stroke-specific glycemic control plans and might also guide tailored treatment approaches depending on the type of stroke.

### 5.1. Inflammatory Infiltration in Different Types of Stroke

The inflammatory response differs significantly between ischemic and hemorrhagic stroke, both in its pattern and the consequences of inflammatory infiltration. In ischemic stroke, the inflammatory response begins within minutes of vascular occlusion, with microglia being the first activated as early as 30 min. Microglia initially express both pro-inflammatory cytokines such as TNF-α and IL-1β, as well as anti-inflammatory mediators such as interleukin-1 receptor antagonist (IL-1Ra) [38]. Neutrophils follow, contributing to BBB disruption, hemorrhagic transformation, and microvascular blockage through the release of reactive oxygen species (ROS), matrix metalloproteinases (MMPs), and proteolytic enzymes [38,39]. Monocytes differentiate into macrophages that may differentiate into either a pro-inflammatory M1 or anti-inflammatory M2 phenotype. T cells also participate, with pro-inflammatory subsets exacerbating injury, while regulatory T cells (Tregs) exert protective effects through interleukin-10 (IL-10) and transforming growth factor beta (TGF-β) [39].

In hemorrhagic stroke, bleeding-related tissue damage initiates inflammation by releasing damage-associated molecular patterns (DAMPs). Moreover, the inflammation is exacerbated by the release of hemoglobin, heme, and iron from lysed red blood cells via oxidative stress. Microglia and macrophages are the first responders, rapidly activated by DAMPs binding to toll-like receptors (TLRs), particularly TLR4. This activation triggers pro-inflammatory gene expression through the nuclear factor kappa-light-chain-enhancer of activated B cells (NF-κB) pathway [39]. Neutrophils infiltrate, releasing MMP-9, myeloperoxidase (MPO), and neutrophil extracellular traps (NETs), which increase BBB permeability, edema, and neuronal injury. Similar to ischemic stroke, monocytes and T cells also contribute to the inflammatory response in hemorrhagic stroke [40].

Cytokine expression in both stroke types follows a biphasic pattern: an early pro-inflammatory phase dominated by TNF-α, IL-6, IL-1β, interleukin-27A (IL-17A), and high-mobility group box 1 (HMGB1), followed by a later anti-inflammatory phase marked by IL-10, TGF-β1, and interleukin-27 (IL-27) [39,40]. Balancing these inflammatory processes is crucial, as excessive or prolonged inflammation worsens neurological outcomes [39].

### 5.2. The Role of SH in Inflammatory Infiltration

SH exacerbates neuroinflammatory processes in both ischemic and hemorrhagic stroke. In acute ischemic stroke, SH induces excessive anaerobic glycolysis, leading to lactate accumulation, acidosis, and increased oxidative stress. This oxidative environment activates NADPH oxidase and damages mitochondria, further disrupting the BBB and promoting leukocyte infiltration. SH promotes neutrophil recruitment and elevates the secretion of pro-inflammatory cytokines, including IL-1β and TNF-α, thereby worsening tissue injury [33].

In hemorrhagic stroke, hyperglycemia accelerates BBB disruption and impairs microvascular integrity, contributing to ongoing bleeding and hematoma expansion. The consequent generation of thrombin triggers inflammatory cascades, elevating white blood cell count, C-reactive protein (CRP), IL-6, TNF-α, and IL-18 levels [34].

## 6. SH as a Prognostic Indicator

SH has either equal or even better predictive power than other prognostic biomarkers in stroke. An indication of systemic inflammation, the neutrophil-to-lymphocyte ratio (NLR) has been shown to correlate with stroke outcomes and infarct volume [41]. Stroke outcomes have also been linked to CRP, troponin, brain natriuretic peptide (BNP), and D-dimer [42]. In a meta-analysis of 70 biomarkers across categories such as inflammatory, hemostatic, glial damage, excitatory neurotransmitter, cardiac, anti-inflammatory, anticlotting, and miscellaneous markers, no single class showed a clearly stronger association with poor outcome than the others. However, cardiac markers, such as troponin and natriuretic peptides, demonstrated a notably consistent link with outcomes [43]. SH, however, presents a unique and all-encompassing biomarker since it combines neuroinflammatory signaling with metabolic stress [42]. It correlates with HPA axis and sympathetic nervous system activation, catecholamine and cortisol surges, and subsequent glucose elevation, all of which can exacerbate secondary brain injury [44].

### 6.1. Short-Term Outcome: Complications, Hospital Stay, and Death

The value of SH as a prognostic indicator of ischemic stroke outcomes is supported by the strong correlation between SH and higher in-hospital mortality, longer length of stay, and higher rates of complications, including urinary tract infections, pressure ulcers, and respiratory infections in ischemic stroke patients [45,46]. Whereas, in hemorrhagic stroke, it has been linked to early adverse effects, including greater hematoma volume and higher in-hospital mortality rate [11,47]. Upon admission, patients with a higher SH index often present with higher NIHSS scores, indicating more severe strokes [48]. Additionally, they are more likely to need intensive care interventions, including mechanical ventilation and vasopressors [46].

The timing of glucose measurement critically influences the prognostic utility of SH. Capes et al. (2001) showed that early hyperglycemia, particularly within 24 h, was a stronger predictor of a threefold increase in short-term mortality and poor functional recovery [49]. Similarly, the ECASS-II trial highlighted that persistent hyperglycemia, defined by elevations at both admission and 24 h, was significantly associated with reduced early neurological improvement and increased 90-day mortality. Notably, even delayed hyperglycemia at 24 h was independently predictive of increased risk of parenchymal hemorrhage and death, suggesting that hyperglycemia at any stage of the acute phase is detrimental [50]. Nonetheless, studies have demonstrated that blood glucose levels tend to decrease during the first 24 h following stroke onset but rise again between approximately 24 to 88 h, likely due to disrupted glucose metabolism that becomes apparent when the patient resumes feeding after an initial period of fasting, reflecting a biphasic glucose pattern [51]. This finding highlights the importance of serial or continuous glucose monitoring over single-point measurements for more accurate prognostic stratification.

Palaiodimou et al. (2021) further supported this concept, showing that increased glycemic variability (GV) during the first 96 h was significantly associated with poorer neurological improvement during hospitalization [52]. The role of stress hormones in this process is also significant. The acute phase of ischemic stroke, extending through the first week, is accompanied by high levels of humoral cortisol and catecholamines, increasing glycogenolysis, gluconeogenesis, proteolysis, and lipolysis, resulting in excessive glucose production, which has been linked to worse outcomes [51].

### 6.2. Long-Term Outcomes: Functional and Motor Recovery and Disability

SH has been shown to compromise functional independence over time, as evidenced by a lower barthel index (BI) at 28-days post-stroke with reduced ability to perform normal activities of daily living (ADLs) [53]. It has been shown that patients with a SHR ≥ 0.96 were significantly more likely to exhibit unfavorable functional outcomes, defined as mRS scores of 3–6, at 3 months following MT [54]. Moreover, patients with high SH scores are less likely to achieve a rapid and substantial motor recovery, as they show consistently low fugl–meyer assessment (FMA) scores and they often suffer from long-term disability and reduced quality of life [55]. Therefore, focused glucose monitoring and control are beneficial for long-term neurological recovery maximization [56,57,58].

## 7. Impact of Chronic Versus Acute Hyperglycemia on Stroke Outcomes

Clinical outcomes in stroke patients are much influenced by differentiating pre-existing diabetes from non-diabetic SH. Patients with pre-existing DM showing elevated glucose levels over time undergo adaptive cellular changes, including upregulation of glucose transporters and improved tolerance to oxidative stress. This adaptation clarifies why diabetic patients—who start with higher glucose levels—sometimes have less severe metabolic disturbances than non-diabetic SH patients [59]. Conversely, non-diabetic SH displays fast dysregulation of glucose metabolism resulting from stress-induced hormone oscillations, which have a more immediate and significant impact on brain metabolism [23]. Due to a lack of metabolic adaptation observed in diabetic patients, non-diabetic patients are more vulnerable to the negative effects of acute hyperglycemia, including increased oxidative stress, excitotoxicity, and BBB disturbance [60].

In those patients, SH appears to have a more marked negative predictive value than in diabetic patients. It has been shown that non-diabetic SH in AIS patients is associated with greater morbidity, poorer functional recovery (Lower BI), greater stroke severity (hHigher NIHSS), and higher infarct volumes than pre-existing DM [48,61,62]. Moreover, they are more prone to recovery delay due to hemorrhagic transformation [63,64]. Non-diabetic patients who experience stroke-related bleeding usually have higher short-term mortality, worse functional outcomes (Lower mRS), and higher stroke severity (Higher NIHSS) [49,56]. These differences underline the need to separate SH from chronic hyperglycemia in clinical practice since treatment plans might be modified according to the baseline metabolic condition of the patient. Additionally, since SH in non-diabetic patients reflects a more pronounced inflammatory or ischemic response, it may warrant tailored glucose control and closer monitoring. However, in a multicenter study involving 1098 patients, admission hyperglycemia was independently associated with increased risks of death, symptomatic intracerebral hemorrhage (sICH), and a decreased probability of favorable functional status at 90 days, regardless of DM status [65]. This suggests that the extent of hyperglycemia at presentation may be a more critical determinant of outcome than whether it arises from chronic diabetes or an acute stress response.

Although severe hypoglycemia in non-diabetic patients should be taken into consideration, glucose overcorrection in both groups may signal more severe underlying pathology than a temporary metabolic reaction. Diabetic and non-diabetic patients show different physiological responses to SH. Persistent hyperglycemia and natural insulin resistance mean that SH may be less apparent in diabetic patients and require higher glucose levels to start. Conversely, non-diabetic individuals are more likely to have significant glucose surges under stress, even with minimal increases in glucose levels, which possibly lead to negative effects [49,66]. This difference determines patient management. Under acute stress, diabetic patients may require more rigorous insulin treatment; non-diabetic patients might just need temporary glucose stabilization during the stress phase [56]. These differences allow physicians to more precisely modify treatment plans.

## 8. Measurement and Assessment of SH

Although there are several SH measurement methods, there is no scientific consensus regarding the best measurement method and the optimal threshold for predicting outcomes following stroke (Table 1). This challenge needs to be addressed, as it represents a difficulty in diagnosing and treating SH in acute settings. Depending on the clinical setting and patient demographic factors, the optimal threshold for defining SH varies considerably in the literature; reported random blood glucose (RBG) thresholds range from >140 mg/dL (7.8 mmol/L) to >180 mg/dL (10 mmol/L) or even >200 mg/dL (11.1 mmol/L) [5,31]. Such variance may be explained by the different physiological responses observed in diabetic compared to non-diabetic patients, the type of the involved stressor, and inconsistencies in diagnostic tools and outcome metrics [67,68,69]. This variability reflects an ongoing controversy in the literature, as no unified standard or guideline exists regarding the most accurate or clinically useful measure of SH in stroke.

For example, El-Gendy et al. (2021) reported that an RBG > 145 mg/dL predicted 30-day mortality and hemorrhagic transformation [46], while Latha V and Shashibhushan J (2024) found that > 200 mg/dL was a significant predictor of poor outcomes [45]. Masrur et al. (2015), however, found that blood glucose > 140 mg/dL in non-diabetic patients and HbA1c > 6.5 in diabetic individuals were linked to worse outcomes, including sICH, life-threatening hemorrhage, in-hospital mortality, and length of stay, without additional poor outcomes above the blood glucose level of 200 mg/dL or HbA1c > 8 [70]. These findings emphasize the limitations of using fixed absolute glucose thresholds and underscore the need for individualized interpretation.

Studies such as that by Capes et al. (2001) showed stronger prognostic associations between acute hyperglycemia and poor outcomes in stroke patients, while diabetic patients exhibit less pronounced effects [49]. Hyperglycemia in diabetic individuals may reflect chronic poor glycemic control rather than acute stress alone [15]. This difference justifies using SH indices (e.g., SHR and GG), which are relative values derived from background blood glucose as a reference, instead of absolute glucose cutoffs [17]. For instance, SHR2 ≥ 0.98, calculated as fasting blood glucose divided by the A1c-derived average glucose (ADAG), was associated with hematoma expansion, neurological deterioration, 30-day mortality, and poor 3-month mRS in hemorrhagic stroke [11]. SHR2 > 0.89 was associated with 3-month poor functional outcomes and hemorrhagic transformation in ischemic stroke. Similarly, the GG > −0.53 as calculated by subtracting ADAG from fasting glucose, which was linked to such adverse outcomes [17]. The glucose-to-A1c ratio (GAR), calculated as fasting plasma glucose divided by HbA1c, has emerged as another marker of relative hyperglycemia. It has been shown to predict stroke recurrence, sICH, poor 3-month outcomes, and 1-year mortality in ischemic stroke patients [16,63]. Such measurement methods should be taken into consideration, especially for diabetic patients where RBG values are not useful.

Although these parameters can be valuable in those with DM, the true blood glucose fluctuations are not accurately captured, which may limit their clinical utility. GV metrics, measured by short term continuous glucose monitoring (CGM), offer another dimension, reflecting fluctuations in glucose rather than absolute levels [52]. Mostly, short term GV is reported thorough standard deviation (SD), mean amplitude of glycemic excursions (MAGE), and coefficient of variation (CoV). High GVs, as measured by SD > 20.9 mg/dL, CoV < 15%, and J-index > 21, have all been associated with poor 3-month functional outcomes [71,72]. MAGE > 5.6 mg/dL and mean absolute glucose (MAG > 19 mg/dL/h) were also predictive of END and poor neurological recovery during hospitalization, respectively [52,73]. Glycated albumin (GA) is an additional indicator of GV that is gaining interest. GA ≥ 16% has been associated with poor outcomes in both ischemic and hemorrhagic stroke [74,75].

Given these discrepancies, there is a need for more reliable, standardized assessment tools that integrate both acute and chronic glycemic data to accurately stratify risk and guide interventions. These refined measures help distinguish true stress-induced hyperglycemia from chronic dysglycemia, particularly in non-diabetic stroke patients who are more vulnerable to glucose-related complications. Relying solely on high absolute glucose thresholds may delay recognition of clinically significant hyperglycemia and hinder timely, targeted interventions in patients at elevated risk of poor outcomes. biomedicines-13-01834-t001_Table 1Table 1Overview of cutoff values for SH in stroke.Measurement MethodIschemic StrokeHemorrhagic StrokeStudyCutoff ValueAssociated OutcomesCutoff ValueAssociated OutcomesRBG, mg/dL>200Poor functional outcomes; prolonged hospital stay; UTI; bedsores; lower respiratory tract infections NANA[45]>14530-day mortality; mechanical ventilation and vasopressors requirement; hemorrhagic transformationNANA[46] 155 Higher admission stroke severity; 3-month poor functional outcomes; 3-month mortality; increased infarct volumeNANA[57] NANA115.2–199.83-month poor clinical outcome[76]FBG, mg/dLNANA111.6–135Poor outcome; DCI[77]ABG, mg/dLNANA>180 Inpatient mortality; unfavorable discharge mRS[78]NANANA30- and 90-day all-cause mortality[79]110–126 In-hospital and 30-day mortality120.6–126Not associated with short-term mortality[49] 130 Higher stroke severity; functional impairment; 90-day mortalityNANA[80] 131.4 12-month poor functional outcomesNANA[81] 140 3-month poor functional outcomes; post-stroke infection; 3-month mortalityNANA[82] 140 3-month poor functional outcomes; 3-month mortality; sICHNANA[83] 140.4 Poor functional outcomes at dischargeNANA[84] ≥140 3-month poor functional outcomes; sICHNANA[85] 140.4 3-month poor functional outcomesNANA[86] 140.4 Not relevant with 3-month poor functional outcomes and sICHNANA[13] 200 7-day mortality; 3-month mortality; ENDNANA[86] GVSD, mg/dL≥20.93-month poor functional outcomesNANA[71] SD, mg/dLNANA15.9–27.4 Unfavorable neurological outcomes[87] SD CoVNA28- and 90-day mortalityNA28- and 90-day mortality[88] CoV, %<153-month poor functional outcomesNANA[72] J index>213-month poor functional outcomesNANA[72] J index≥6.553-month cardiovascular outcomes NANA[89] MAGNAPSCINANA[90] MAG>19Poor neurological improvement during hospitalization>19Poor neurological improvement during hospitalization[52] MAGE>5.6ENDNANA[73] TRNA3-month poor functional outcomes; sICHNANA[91] SHRSHR1SHR2SHR3≥0.98Poor functional outcome; hemorrhagic transformationNANA[62] SHR1≥27.59Higher stroke severity; poor neurological statusNANA[92] SHR2≥0.973-month poor functional outcomesNANA[93] SHR2NANA≥0.98Hematoma expansion; neurological deterioration; 30-day mortality; 3-month poor mRS[11] SHR2>0.893-month poor functional outcomes; hemorrhagic transformationNANA[17] SHR2≥1.0712-month stroke recurrence; 12-month all-cause deathNANA[63] SHR2NA12-month severe neurological deficit; 12-month mortality NANA[60] SHR2NAHemorrhagic transformationNANA[94] SHR2>1.32In-hospital mortalityNANA[18] SHR2NAEND; poor functional outcomes at dischargeNANA[95] SHR2NANANAIn-hospital mortality; hematoma expansion[96] SHR3≥1.14In-hospital mortality; poor functional outcomes; stroke extension; hemorrhagic transformationNANA[61] SHR3≥0.963-month poor functional outcomesNANA[54] SHR3NANANADCI; 90-days poor prognosis[76] SHR3NANANA30-day and 1-year mortality[97] SHR3>0.81In-hospital mortality; prolonged hospital stays>81In-hospital mortality; prolonged hospital stays[98] GA, %NANA≥16.0Hematoma expansion; END;1-month mortality; and 3-month poor functional outcomes[75] ≥16Unfavorable short-term outcomesNANA[74]≥16END

[99] NA3-month and 1-year poor functional outcomes; 1-year stroke recurrence; 1-year combined vascular events NANA[100]GG>−0.533-month poor functional outcomes; hemorrhagic transformation NANA[17] 45 mg/dlHigher stroke severity; poor neurological statusNANA[92] GARNAStroke recurrence, 1-year mortality NANA[63] NA3-month poor outcomes; 3-month mortality; sICHNANA[47] NA3-month poor outcomes; 3-month mortality; sICHNANA[16] NANA≥1.02Poor functional outcome[101] HbA1c, %NANA≥6.0Unfavorable clinical outcomes[102]5.4–7.1Reduced functional independence; mortality; sICHNANA[103] >8.2PSCINANA[104] NA: Not applicable; RBG: Random blood glucose; FBG: Fasting blood glucose; DCI: Delayed cerebrovascular ischemia; ABG: Admission blood glucose; mRS: Modified rankin scale; sICH: Symptomatic intracerebral hemorrhage; END: Early neurological deterioration; GV: Glucose variability; SD: Standard deviation; CoV: Coefficient of variance; MAG: Mean absolute glucose; PSCI: Post-stroke cognitive impairment; MAGE: Mean amplitude of glycemic excursions; TR: Time rate; SHR: Stress hyperglycemia ratio; GA: Glycated albumin; GG: Glycemic gap; GAR; Glucose-to-A1c ratio; J index: 0.001(mean blood glucose + SD)^2^ for glucose measured in mg/d; SHR1: Fasting glucose (mmol/L)/HbA1c (%); SHR2: Fasting glucose (mmol/L)/[(1.59 × HbA1c) − 2.59]; SHR3: Admission blood glucose (mmol/L)/[(1.59 × HbA1c) − 2.59]; GG: fFasting blood glucose (mmol/L) − [28.7 × HbA1c) − 46.7]; GAR: Fasting plasma glucose (mg/dl)/HbA1c (%).

## 9. Management of SH in Stroke

### 9.1. Management Outline

The management of SH in stroke patients is guided by current clinical recommendations that aim to optimize glycemic control while minimizing the risks associated with both hyperglycemia and hypoglycemia [18]. The AHA and the American diabetes association (ADA) recommend maintaining blood glucose levels between 140 and 180 mg/dL (7.8–10 mmol/L) in acute stroke patients to reduce the risk of adverse outcomes, including reducing mortality and neurological complications [105]. Similarly, the national institute for health and care excellence (NICE) guideline NG128 advises maintaining glucose levels between 4 and 11 mmol/L (72–198 mg/dL) [106]. The 2020 Canadian stroke best practice recommendations (CSBPR) emphasize the need for individualized glycemic targets and warn against both hypoglycemia and hyperglycemia. They recommend maintaining an HbA1c of ≤7.0%, fasting plasma glucose levels between 4.0 and 7.0 mmol/L, and 2-h postprandial glucose levels within the range of 5.0 to 10.0 mmol/L [107]. It has been shown that lower glycemic targets are essential for SH in non-diabetic patients (less than 7.8 mmol/L or 140 mg/dL) [108].

Importantly, in patients receiving reperfusion therapies such as intravenous thrombolysis (IVT) with alteplase or MT, glucose levels have a significant impact on reperfusion injury and hemorrhagic transformation [109,110]. In the 2013 guidelines from the American heart association/American stroke association (AHA/ASA), eligibility for alteplase administration required blood glucose levels to fall between 50 and 400 mg/dL, a criterion originally based on the inclusion parameters of the NINDS tPA Trial. However, the recent AHA/ASA guidelines have updated this recommendation, advising exclusion only for patients with glucose levels below 50 mg/dL (Class I; Level of evidence A). For patients presenting with blood glucose levels above 400 mg/dL, intravenous alteplase may still be considered if glucose levels are promptly corrected using appropriate treatment (e.g., Insulin or Dextrose) and are followed by a neurological reassessment within approximately 15 min [110]. Data from the safe implementation of treatments in stroke–international stroke thrombolysis register (SITS-ISTR) also indicate that blood glucose levels exceeding 180 mg/dL at the time of thrombolysis should be avoided, as they are associated with an increased risk of sICH [111]. For MT, there is currently no evidence to suggest that hyperglycemia should be considered a contraindication, and the AHA guidelines still recommend treating hyperglycemia to achieve target glucose levels of 140 and 180 mg/dL (7.7–10.0 mmol/L), due to a lack of clinical trials [109]. Conversely, current joint commission standards for the certification of primary stroke centers do not require specific protocols for hyperglycemia management. As a result, most centers continue to rely on standard sliding-scale insulin regimens [112].

### 9.2. Differences Between SH Management in Ischemic Versus Hemorrhagic Stroke

Glucose reduction criteria vary between ischemic and hemorrhagic strokes due to their distinct pathophysiological mechanisms. In ischemic stroke, glucose-lowering therapy is typically initiated within the first 24 h of admission, especially in patients with severe hyperglycemia (≥180 mg/dL), while avoiding rapid glucose fluctuations that could impair cerebral autoregulation [105]. Gentile et al. (2005) found that the normalization of blood glucose to below 130 mg/dL provided a significant survival advantage and a 4.6-fold reduction in mortality, underscoring the importance of early glycemic control in the management of acute ischemic stroke [113]. For hemorrhagic stroke, tighter glycemic targets in the range of 110–160 mg/dL are often pursued to reduce the risk of hematoma expansion [114]. The intensive care bundle with blood pressure reduction in the acute cerebral hemorrhage (INTERACT-3) trial implemented a care bundle for ICH that included intensive glucose control (110–140 mg/dL for non-diabetic patients, 140–180 mg/dL for diabetic patients), early blood pressure lowering, antipyretics, and anticoagulation reversal, resulting in reduced mortality and poor functional outcomes [115].

### 9.3. Effect of DM Status on SH Management in Stroke

Differences in glycemic targets must also consider the presence or absence of DM. Baird et al. (2003) demonstrated that persistent hyperglycemia in non-diabetic ischemic stroke patients was independently associated with infarct growth and worse recovery [58]. Conversely, in patients with known DM, higher glucose levels may reflect chronic adaptation, and rapid normalization can precipitate cerebral energy crises due to downregulated glucose transport and altered neurovascular coupling [116]. For these reasons, a more aggressive approach is often used in non-diabetic individuals, while glucose targets in patients with DM are maintained closer to their baseline levels to avoid harmful fluctuations. Supporting this, in the hyperglycemia insulin network effort (SHINE) trial, 80% of the patients had a history of diabetes, making the findings particularly relevant to diabetic individuals with acute ischemic stroke. This trial found that intensive glucose treatment targeting 80–130 mg/dL experienced significantly more hypoglycemia, primarily among diabetic patients, suggesting that aggressive normalization may be harmful in this subgroup [117]. Similarly, in the glucose regulation in acute stroke patients (GRASP) trial, where 59% of participants had DM, the tight control group that targeted 70–110 mg/dL experienced a 30% incidence of hypoglycemia, further highlighting the risks associated with overly strict glucose targets in diabetic stroke patients [118].

### 9.4. Risk of Hypoglycemia

Maintaining optimal glucose levels in stroke patients is challenging, as both hyperglycemia and hypoglycemia carry risks. The INSULINFARCT trial found a significantly higher rate of hypoglycemia in the intensive insulin therapy (IIT) group—5.7% (<54 mg/dL) and 34.5% (<64 mg/dL)—compared to controls. Notably, infarct growth and extracranial hemorrhagic transformation were significantly higher in the IIT group, suggesting a potential link between hypoglycemia and adverse outcomes [119]. Hypoglycemia was attributed to exacerbating stroke severity and increasing the risk of hemorrhagic transformation by triggering inflammatory responses, disrupting the BBB, inducing acute hypertension, and altering platelet function [119]. However, the treatment of hyperglycemia in the ischemic stroke (THIS) trial and the GRASP trial did not find a significant correlation between hypoglycemia (<60 mg/dL and <55 mg/dL, respectively) and worsened clinical outcomes [118,120]. This discrepancy may have resulted from using different definitions of hypoglycemia across trials, making it difficult to draw uniform conclusions about thresholds for harm.

### 9.5. Treatment Options for SH in Stroke

The cornerstone of glucose management in acute stroke remains insulin therapy [121]. Beyond glycemic regulation, insulin has neuroprotective properties, including anti-inflammatory effects, suppression of oxidative stress, improved endothelial function, and modulation of cerebral vasodilation [122]. Intravenous insulin infusions are preferred in intensive care settings, particularly for patients with persistent glucose > 180 mg/dL [123], to rapidly correct hyperglycemia and maintain stable glucose levels [122]. While subcutaneous insulin regimens are employed in more stable patients [123]. The optimal timing for initiating insulin therapy in acute stroke remains uncertain; however, early initiation, ideally within the first 24 to 48 h of hospital admission [108], is considered beneficial and is supported by AHA recommendations [109]. Studies such as those by Bruno et al. (2004) [124] and Johnston et al. (2019) [117] initiated insulin infusions within the first 12 h after stroke onset, aiming to begin before significant ischemic damage developed and to target early harmful cellular mechanisms. Regarding the duration of insulin therapy, established guideline recommendations are lacking. However, for subcutaneous insulin, it is generally recommended to continue the regimen from inpatient to outpatient care, whereas intravenous insulin therapy is typically considered most effective when administered for approximately 48 h [121]. Walters et al. (2006) found that extending insulin infusion beyond 48 h did not yield additional benefits, supporting this time frame as optimal [125]. However, insulin therapy requires frequent monitoring and poses a significant risk of hypoglycemia [126].

Emerging strategies include CGM systems to track glycemic fluctuations in real time and individualized glucose control protocols, minimizing both hypo- and hyperglycemia [123]. The endocrine society clinical practice guideline recommends that for hospitalized adults with insulin-treated DM and noncritical illness who are at high risk for hypoglycemia, real-time CGM should be used alongside confirmatory bedside point-of-care (POC) blood glucose testing to guide insulin dosing, instead of relying solely on POC testing [127].

Large trials have failed to support intensive glycemic control in stroke. The SHINE trial demonstrated that intensive glucose control targeting 80–130 mg/dL via continuous insulin infusion did not result in a significant improvement in 90-day functional outcomes compared to standard subcutaneous insulin therapy and was associated with a markedly higher incidence of severe hypoglycemia (2.6% vs. 0%) [117]. Southerland et al. (2024), in a prespecified analysis of the SHINE trial, reported that IIT did not lower the risk of sICH following thrombolysis when compared to standard sliding scale insulin management [128]. The UK glucose insulin in stroke trial (GIST-UK) also failed to demonstrate a benefit in all-cause mortality and death or disability at 90 days with glucose-potassium-insulin (GKI) infusion in hyperglycemic patients [129]. A meta-analysis also concluded that tight glucose control (<135 mg/dL) using insulin infusion does not offer clinical benefits and significantly increases the risk of hypoglycemic episodes [130]. These outcomes may be partly explained by the broad and heterogeneous patient populations enrolled in these trials, including a high proportion of patients with mild to moderate stroke severity, in which natural recovery may overshadow potential treatment effects. Furthermore, the higher frequency of severe hypoglycemia and treatment discontinuation observed in the intensive therapy group was likely counterbalanced by theoretical neuroprotective benefits of tight glucose control [117].

Although insulin remains the primary treatment [122], some oral glycemic agents have shown potential neuroprotective benefits beyond glucose control; however, they are not recommended for managing post-stroke hyperglycemia due to the increased risk of hypoglycemia and acidosis in acutely ill patients, particularly in dysphagic stroke patients [108]. Glucagon-like peptide-1 receptor agonists (GLP-1 RAs) promote insulin secretion and have been shown in preclinical studies to reduce infarct volume, oxidative stress, and BBB permeability, while enhancing angiogenesis and neurogenesis [131]. Dipeptidyl peptidase-4 (DPP-4) inhibitors increase endogenous GLP-1 levels and may preserve cerebrovascular integrity by activating the PI3K/AKT and mTOR pathways, thereby improving endothelial function and reducing inflammation [132]. Additionally, experimental models suggest that sodium-glucose cotransporter-2 (SGLT2) inhibitors may reduce neuroinflammation and plaque size, offering potential roles in the secondary prevention of stroke [133]. The 2020 update of the Canadian stroke best practice recommendations CSBPR for secondary prevention of stroke advises that, for patients with stroke and type 2 DM who do not reach glycemic targets with standard oral medications, antihyperglycemic agents shown to improve major cardiovascular outcomes, such as SGLT-2 inhibitors or GLP-1 RAs, should be considered as part of the treatment strategy [107]. Other glucose-lowering agents are being explored for their potential roles in stroke care. Metformin has shown promise in the subacute and chronic phases of stroke recovery due to its ability to enhance endothelial nitric oxide synthase, stimulate neurogenesis, reduce oxidative stress, and inhibit apoptosis [134]. In addition, thiazolidinediones were found to reduce inflammation and infarct volume [102] (Table 2).

Furthermore, emerging evidence suggests that several natural products exhibit neuroprotective effects in hyperglycemia-exacerbated stroke models. In a mouse model of ischemic stroke with acute hyperglycemia, lauric acid (LA) significantly attenuated infarct volume and brain edema while enhancing the expression of tight junction proteins and antioxidative markers, as well as maintaining BBB integrity. LA also improved motor coordination, survival, and nutritional status, demonstrating comprehensive neuroprotection [135]. Similarly, the salvianolate lyophilized injection (SLI), composed of active components from Salvia Miltiorrhiza, has been shown to protect against cerebral ischemia in type 1 diabetic rats by increasing brain microvasculature, enhancing glucose uptake, and reducing inflammation [136]. Additionally, baicalin, a flavonoid derived from Scutellaria Baicalensis, has shown protective effects by regulating mitochondrial dynamics, reducing ROS production, enhancing mitophagy, and preserving mitochondrial membrane potential through AMP-activated protein kinase (AMPK)-dependent pathways [137]. Although these natural compounds show promise, further clinical research is warranted to determine their safety and efficacy in stroke patients with SH. biomedicines-13-01834-t002_Table 2Table 2SH management options in stroke.Management OptionHypoglycemic EffectNeuroprotective EffectStudyInsulin Reduction in gluconeogenesis; increased glucose uptake through GLUT4 transporters; glycogenesisSuppression of neuroinflammation, ROS formation, lipolysis, and platelet aggregation; vasodilation [122,138]SGLT2 InhibitorInhibiting renal SGLT2 transporter that decrease renal glucose re-absorption, increasing urinary glucose excretionReduction in neuroinflammation and plaque size[133,139]GLP-1 Receptor AgonistsIncreased glucose-induced insulin secretion; glycogenesis Reduction in infarct volume, apoptosis, oxidative stress, neuroinflammation, excitotoxicity, and BBB permeability; increased neurogenesis, neuroplasticity, angiogenesis, and cerebral perfusion[131]DPP-4 inhibitors Enhanced bioavailability of GLP-1 and GIP by Inhibition of DPP-4 enzymeActivation of the AKT/MTOR pathway; suppression of BBB disruption, oxidative stress, apoptosis, and inflammation; enhanced endothelium relaxation and cerebrovascular remodeling[132,140]MetforminReduction in gluconeogenesis; increased insulin receptor sensitivity; decreased glucose uptake in the intestineSuppression of oxidative stress and apoptosis; endothelial nitric oxide synthase activation; activation of angiogenesis and neurogenesis[134] ThiazolidinedionesEnhance insulin sensitivity and reduce serum glucose by PPAR-γ activationReduce neuroinflammation and infarct volume[102] α-Glucosidase inhibitorsDelayed carbohydrate absorption from intestine by inhibiting α-glucosidase enzymesInhibitory potential on DAPK1-p53 interaction; prevention of mitochondrial and lysosomal dysfunction; favorable modulation of gene expression related to cell survival, inflammation, and regeneration[141,142]GLUT4: Glucose transporter type 4; ROS: Reactive oxygen species; SGLT2: Sodium-glucose cotransporter 2; GLP-1: Glucagon-like peptide-1; DPP-4: Dipeptidyl peptidase-4; GIP: Glucose-dependent insulinotropic polypeptide; AKT: Protein kinase B; MTOR: Mechanistic target of rapamycin; BBB: Blood–brain barrier; PPAR-γ: Peroxisome proliferator-activated receptor gamma; DAPK1: Death-associated protein kinase 1.

## 10. Study Strengths and Limitations

To the best of our knowledge, our study represents the most comprehensive study that analyzes SH value in both hemorrhagic and ischemic stroke. Despite its comprehensiveness, several limitations should be considered. Stroke often coexists with other vascular risk factors, including DM, HTN, dyslipidemia, and smoking. Such overlap makes it difficult to define SH’s independent prognostic value in stroke outcomes. Furthermore, subclinical DM may complicate the differentiation between hyperglycemia as a stress reaction and as a sign of chronic metabolic dysfunction. The inconsistent SH measurement methods further complicate and dilute the withdrawn conclusion’s solidity and make it difficult to identify which is the most reliable measurement technique. All of these, combined with the scarcity of SH assessments in hemorrhagic patients, make it difficult to generalize the drawn conclusion.

## 11. Future Directions

The implementation of randomized controlled trials with longitudinal follow-up periods is paramount. Especially when it comes to the hemorrhagic stroke population, there is less focus on such a devastating condition. Although there is a special interest in SH among ischemic stroke patients, there is no clear delineation of SH predictive outcome in different ischemic stroke subtypes. Addressing such a gap may improve the precision of risk stratification and tailor individualized management regimens. Furthermore, the integration of SH into routine stroke assessment models can be a valuable tool to optimize patients, facilitate early intervention, and ultimately improve functional outcomes and recovery trajectories. International collaboration and the adoption of a definite and consistent SH assessment tool are crucial in enhancing predictive accuracy and advanced routine clinical application.

## 12. Conclusions

Recently, SH has become a vital and consistent predictor of adverse clinical outcomes in acute stroke patients. Higher risks of mortality, poor neurological recovery, hemorrhagic transformation, and larger infarct volumes are all strongly correlated with increasing SH levels. Patients without a history of DM show such effects more clearly, suggesting that acute stress-induced glucose increments may have a special role in stroke pathogenesis. The constant association between SH and stroke outcomes emphasizes the potential need to include SH assessments in standard clinical practice. Integrating routine SH evaluation at admission—as a prognostic marker alongside established tools such as the NIHSS and Alberta stroke program early CT score (ASPECTS)—can improve risk stratification and clinical decision-making. Using SH-guided approaches could help customize glycemic treatments, improving patient outcomes and lowering the risks connected to both hyperglycemia and hypoglycemia. Early SH identification and management may enhance the quality of stroke treatment, raise prognostic accuracy, and enable quick intervention. Regular SH monitoring also aids in predicting complications such as hemorrhagic transformation and cerebral edema, informing critical decisions regarding thrombolysis or neurosurgical procedures. As our understanding of SH continues to evolve, incorporating its assessment into stroke treatment pathways may promote more individualized, effective, and patient-centered care protocols.

## Figures and Tables

**Figure 1 biomedicines-13-01834-f001:**
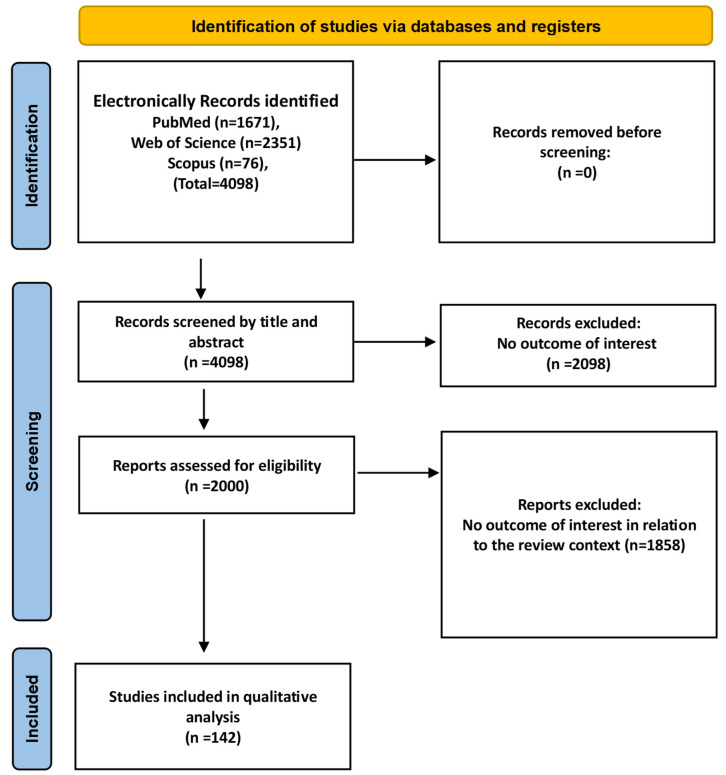
PRISMA flowchart of the identification of the included studies.

**Figure 2 biomedicines-13-01834-f002:**
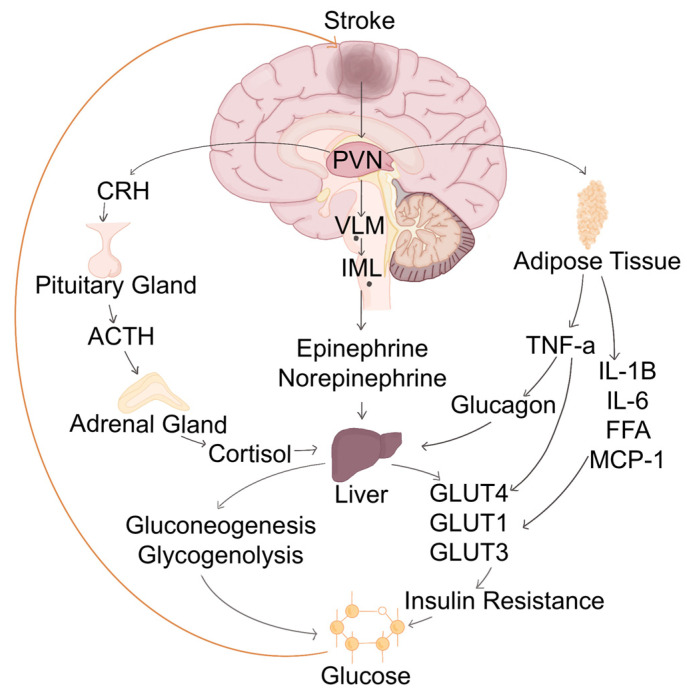
Pathophysiology of SH in stroke. SH mechanisms represent a complex interaction between glucose regulatory centers, the HPA axis, the sympathetic adrenomedullary system, and various humoral factors. Activation of the VMH by stress signals will initiate excessive glucose production via stimulation of the HPA axis and the sympathetic adrenomedullary system. The adrenal medulla is stimulated to release catecholamines through the paraventricular nucleus (PVN)–ventrolateral medulla (VLM)–intermediolateral column (IML) pathway. Concurrently, the adrenal cortex is stimulated to release cortisol. These hormones, together with the effects of glucagon, and growth hormone promote hepatic glycogenolysis and gluconeogenesis. Finally, this response is amplified by the release of pro-inflammatory cytokines, creating a viscous cycle. TNF-a: Tumor necrosis factor-alpha; ACTH: Adrenocorticotropic hormone; CRH: Corticotropin-releasing hormone; IL-1B: Interleukin-1 beta; IL-6: Interleukin-6; FFA: Free fatty acids; MCP-1: Monocyte chemoattractant protein-1; GLUT1: Glucose transporter type 1; GLUT3: Glucose transporter type 3; GLUT4: Glucose transporter type 4.

**Figure 3 biomedicines-13-01834-f003:**
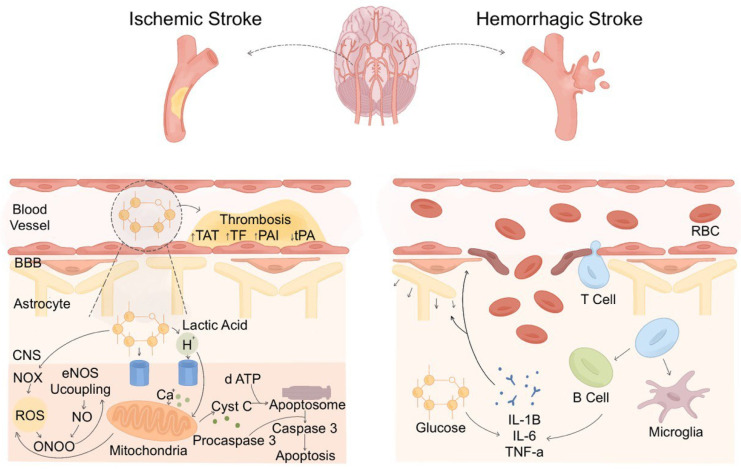
Differences in SH mechanisms between ischemic and hemorrhagic stroke. The mechanism of SH differs in ischemic and hemorrhagic stroke. In ischemic stroke, reduced cerebral perfusion leads to glucose accumulation, promoting thrombosis through increased hyperfibrinolysis via thrombin-antithrombin complex (TAT), tissue factor (TF), plasminogen activator inhibitor (PAI), and reduced tissue plasminogen activator (tPA), which inhibits hyperfibrinolysis. Excess glucose metabolism generates lactic acid, reactive oxygen species (ROS), and peroxynitrite (ONOO^−^) via NADPH oxidase (NOX) and endothelial nitric oxide synthase (eNOS) uncoupling, causing mitochondrial dysfunction, calcium overload, and adenosine triphosphate (ATP) depletion. This activates the apoptotic cascade, including cytochrome c release, apoptosome formation, and caspase-3 activation. In hemorrhagic stroke, vessel rupture leads to BBB breakdown, allowing blood components and immune cells to infiltrate the brain parenchyma. This activates an inflammatory response, with increased release of IL-1, IL-6, and TNF-a, exacerbating secondary injury.

## Data Availability

Not applicable.

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
