# Peer review of "Stress Hyperglycemia as a Prognostic Indicator of the Clinical Outcomes in Patients with Stroke: A Comprehensive Literature Review"

_biomedicines, 2025, doi:10.3390/biomedicines13081834_

Round 1
Reviewer 1 Report
Comments and Suggestions for Authors
This manuscript nicely reviews stress hyperglycemia (SH) in patients with stroke, thus potentially benefiting future studies. I only have some minor suggestions prior to its publication.
Concerning the second paragraph under Section 2, were reviews excluded from consideration? If not, would the authors mind sharing the reason why? Also, for our readers to get a general feeling about the review process, it might be helpful to include the number of literature results obtained at each step.
Concerning Section 9.1, just a quick clarification question. Do the discussed glucose levels refer to long-term management targets, in other words, targets after discharge from the hospital?
Concerning the paragraph starting with “[l]arge trials have failed to support intensive glycemic control in stroke,” is it possible to squeeze in one or two sentences quickly proposing possible reasons behind these failures?
While I was reading the manuscript, some very minor formality matters were noticed, including uses of capital font (e.g., the third paragraph under Section 3, and Tables 1-3), providing full names of acronyms upon their first appearance (e.g., AIS, BI, UTI, FBG, ABG, and CGM), and uses of brackets (e.g., “the (GG)” “the (GAR)” and “MAGE >5.6 mg/dL)”).
Author Response
Comment 1: [Concerning the second paragraph under Section 2, were reviews excluded from consideration? If not, would the authors mind sharing the reason why? Also, for our readers to get a general feeling about the review process, it might be helpful to include the number of literature results obtained at each step.]
Response 1: [Thank you for your valuable and constructive feedback. Please be advised that the exclusion criteria were mentioned on page 4, lines 139-140. Reviews were not excluded because this study adopts a narrative review style to provide comprehensive coverage of SH in the current literature. The flowchart is provided on page 5, Figure 1.]
Comment 2: [Concerning Section 9.1, just a quick clarification question. Do the discussed glucose levels refer to long-term management targets, in other words, targets after discharge from the hospital?]
Response 2: [Thank you for your valuable and constructive feedback. Please be advised that the referred glucose levels pertain to acute stroke management, as mentioned in the same section on page 16, line 478.]
Comment 3: [Concerning the paragraph starting with “[l]arge trials have failed to support intensive glycemic control in stroke,” is it possible to squeeze in one or two sentences quickly proposing possible reasons behind these failures?]
Response 3: [Thank you for your valuable and constructive feedback. Please find the addressed comment on page 18, lines 596–601.]
Comment 4: [While I was reading the manuscript, some very minor formality matters were noticed, including uses of capital font (e.g., the third paragraph under Section 3, and Tables 1-3), providing full names of acronyms upon their first appearance (e.g., AIS, BI, UTI, FBG, ABG, and CGM), and uses of brackets (e.g., “the (GG)” “the (GAR)” and “MAGE >5.6 mg/dL)”).]
Response 4: [Thank you for your valuable and constructive feedback. The acronyms have been revised to ensure that the full names are mentioned at their first appearance. Capitalization has been applied only at the beginning of sentences for the full names of acronyms.]
To ensure reliable conclusions based on your valuable comments, the references have been thoroughly revised throughout the manuscript.
Reviewer 2 Report
Comments and Suggestions for Authors
In this study, the authors provide a comprehensive systematic review of the prognostic value of SH in both ischemic and hemorrhagic stroke, addressing a critical gap in understanding SH’s role in non-diabetic populations. Generally, the review is novel and well-writtened, however, some points still need to be stressed.
- As a systematic review, it should be registered on the PROSPERO, please modify the review according to its requirement.
- In the section SH Prevalence and Risk Factors, the authors claimed "The superimposi-
tion of SH in stroke patient with DM was found to increase the risk of mortality during
hospitalization.", which is inconsistent with the previous statement "non-diabetic patients with SH had a 1.7-fold higher risk of in-hospital 28-day mortality than diabetic patients". It makes it confusing whether DM is a protective factor or risk factor? Please reorganize the content. - The author mentioned the possible mechanism of SH causing poor prognosis of stroke in the review. However, the discussion of the mechanism is not comprehensive and in-depth enough. I suggest that the author expand a subheading to describe in detail: 1. Inflammatory infiltration in different types of stroke; 2. The role of SH in inflammatory infiltration.
- The authors should acknowledge the controversies in SH measurement.
- It is recommended to supplement the author team's unique insights into the SH mechanism (such as the interaction of specific genotypes or biomarkers).
- A clearer answer is needed to the question “How does SH change clinical practice?”
- Claiming that SH is a “strong predictor” requires caution, as the existing evidence is mostly observational studies and causality has not been established.
Author Response
Comment 1: [As a systematic review, it should be registered on the PROSPERO, please modify the review according to its requirement.]
Response 1: [Thank you for your valuable and constructive feedback. Please be advised that this is a narrative review and therefore cannot be submitted to PROSPERO.]
Comment 2: [In the section SH Prevalence and Risk Factors, the authors claimed "The superimposition of SH in stroke patient with DM was found to increase the risk of mortality during hospitalization.", which is inconsistent with the previous statement "non-diabetic patients with SH had a 1.7-fold higher risk of in-hospital 28-day mortality than diabetic patients". It makes it confusing whether DM is a protective factor or risk factor? Please reorganize the content.]
Response 2: [Thank you for your valuable and constructive feedback. There is an ongoing controversy on this matter, as some studies report that diabetic patients experience greater neurological deterioration, while others suggest that non-diabetic patients are more affected. To avoid confusing readers, we have clarified this issue in the revised manuscript on pages 5–6, lines 147–177.]
Comment 3: [The author mentioned the possible mechanism of SH causing poor prognosis of stroke in the review. However, the discussion of the mechanism is not comprehensive and in-depth enough. I suggest that the author expand a subheading to describe in detail: 1. Inflammatory infiltration in different types of stroke; 2. The role of SH in inflammatory infiltration.]
Response 3: [Thank you for your valuable and constructive feedback. We have addressed this issue in the revised manuscript on page 5, lines 250–291.]
Comment 4: [The authors should acknowledge the controversies in SH measurement.]
Response 4: [Thank you for your valuable and constructive feedback. This point has been acknowledged in the revised manuscript on page 12, lines 409–411 and 418–420.]
Comment 5: [It is recommended to supplement the author team's unique insights into the SH mechanism (such as the interaction of specific genotypes or biomarkers).]
Response 5: [Thank you for your valuable and constructive feedback. Unfortunately, there is not enough supportive data available at this time to provide a unique perspective on this matter, as not even theoretical evidence currently exists to address this issue.]
Comment 6: [A clearer answer is needed to the question “How does SH change clinical practice?”]
Response 6: [Thank you for your valuable and constructive feedback. Please find the addressed comment in the revised manuscript on page 21, lines 684–696.]
Comment 7: [Claiming that SH is a “strong predictor” requires caution, as the existing evidence is mostly observational studies and causality has not been established.]
Response 7: [Thank you for your valuable and constructive feedback. Please find the addressed comment in the revised manuscript on page 1, line 36]
To ensure reliable conclusions based on your valuable comments, the references have been thoroughly revised throughout the manuscript.
Reviewer 3 Report
Comments and Suggestions for Authors
This review is important in the field and includes valuable informations. Please respond to the following comments:
1) Numerous studies have been conducted on natural products. They were neglected in this review. Kindly refer to it.
2) The abstract is long. A summarization is required.
3) The content and abbreviation should be explained in Figure 1 caption.
4) Update the reference of α-glucosidase inhibitors in Table 3.
Author Response
Comment 1 [Numerous studies have been conducted on natural products. They were neglected in this review. Kindly refer to it.]
Response 1: [ Thank you for your valuable and constructive feedback. We have addressed this issue accordingly page 19, lines 634–649, last paragraph]
Comment 2: [The abstract is long. A summarization is required]
Response2: [Thank you for your valuable and constructive feedback. The abstract has been revised and summarized accordingly.]
Comment 3: [The content and abbreviation should be explained in Figure 1 caption.]
Response 3: [Thank you for your valuable and constructive feedback. We have addressed this issue accordingly page 7, lines 216–228]
Comment 4: [Update the reference of α-glucosidase inhibitors in Table 3.]
Response 4: [Thank you for your valuable and constructive feedback. We have addressed this issue accordingly (pages 19–20, last cell). The references included were selected based on their relevance to the topic of interest and therefore can not been changed further. Additionally, the previous Table 1 and Figure 3 have been removed to avoid confusion for readers.]
To ensure reliable conclusions based on your valuable comments, the references have been thoroughly revised throughout the manuscript.
Round 2
Reviewer 3 Report
Comments and Suggestions for Authors
No further comments.
The authors provided a good response.